# ADVERSARIAL FILTERS OF DATASET BIASES

## ABSTRACT

Large-scale benchmark datasets have been among the major driving forces in AI, supporting training of models and measuring their progress. The key assumption is that these benchmarks are realistic approximations of the target tasks in the real world. However, while machine performance on these benchmarks advances rapidly — often surpassing human performance — it still struggles on the target tasks in the wild. This raises an important question: whether the surreal high performance on existing benchmarks are inflated due to spurious biases in them, and if so, how we can effectively revise these benchmarks to better simulate more realistic problem distributions in the real world.

In this paper, we posit that while the real world problems consist of a great deal of long-tail problems, existing benchmarks are overly populated with a great deal of similar (thus *non*-tail) problems, which in turn, leads to a major overestimation of true AI performance. To address this challenge, we present a novel framework of *Adversarial Filters* to investigate model-based reduction of dataset biases. We discuss that the optimum bias reduction via AFOPTIMUM is intractable, thus propose AFLITE, an iterative greedy algorithm that adversarially filters out data points to identify a reduced dataset with more realistic problem distributions and considerably less spurious biases.

AFLITE is lightweight and can in principle be applied to any task and dataset. We apply it to popular benchmarks that are practically solved — ImageNet and Natural Language Inference (SNLI, MNLI, QNLI) — and present filtered counterparts as new challenge datasets where the model performance drops considerably (e.g., from 84% to 24% for ImageNet and from 92% to 62% for SNLI), while human performance remains high. An extensive suite of analysis demonstrates that AFLITE effectively reduces measurable dataset biases in both the synthetic and real datasets. Finally, we introduce new measures of dataset biases based on K-nearest-neighbors to help guide future research on dataset developments and bias reduction.

## 1 INTRODUCTION

Large-scale neural networks have achieved superhuman performance across many popular AI benchmarks, for tasks as diverse as image recognition (ImageNet; Russakovsky et al. (2015)), natural language inference (SNLI; Bowman et al. (2015)), and question answering (SQuAD; Rajpurkar et al. (2016)). Yet these deep models struggle when taken out of these dataset environments and evaluated on adversarial data or problems in the wild (Eykholt et al., 2018; Jia & Liang, 2017). This raises a key question: Does high model performance on today's benchmark datasets indicate the underlying *task* is solved, or do those datasets *overestimate the true capabilities of current AI systems*?

Answering this question is key because benchmarks serve important roles in the community. Not only do they *direct* progress on core tasks, they also make it easier to tackle the lofty target tasks such as image recognition in the wild through a more practically-scoped dataset such as ImageNet. However, the closed-world assumption of most existing datasets is subject to significant bias (Torralba & Efros, 2011). Much of the data that is *easy to obtain and label* isn't necessarily representative of the task we seek to measure. Thus, if left unchecked, artifacts from data collection (Fouhey et al., 2018) or human labeling (Gururangan et al., 2018; Poliak et al., 2018; Tsuchiya, 2018; Geva et al., 2019) can significantly inflate model performance. Though there exist task- and dataset-specific ap-

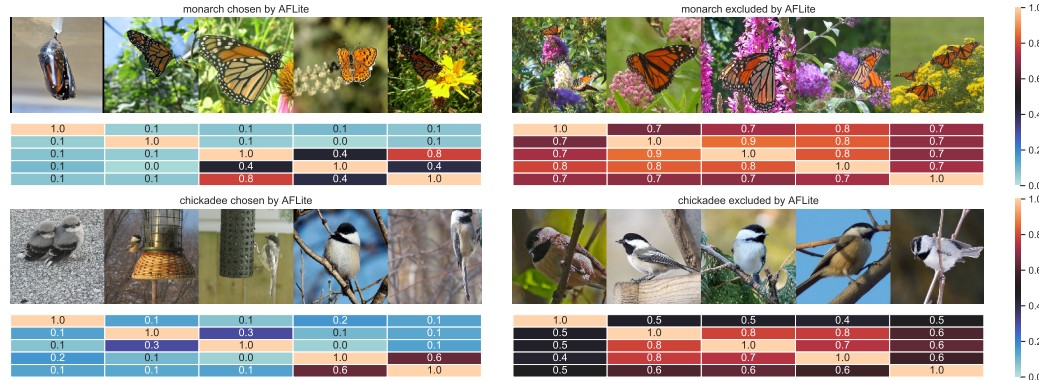

Figure 1: Random ImageNet images for two labels – Monarch Butterfly and Chickadee – that were either selected (left) as adversarial by our AFLITE algorithm, or excluded (right). The heatmap shows pairwise cosine similarity between EfficientNet-B7 features (Tan & Le, 2019). The AFLITE images show significantly greater diversity – such as the cocoon of a butterfly, or the non-canonical chickadee poses – that is in turn reflected by the cosine similarity. This diversity suggests that the AFLITE examples more directly measure progress on the true task of image classification, versus fitting to dataset bias.

proaches for addressing these biases (Goyal et al., 2017; Geirhos et al., 2018), the complex artifacts that emerge from large-scale dataset creation are challenging to exhaustively identify and remove.

In this paper, we present AFLITE – a computationally efficient dataset reduction algorithm, aimed at systematically reducing spurious artifacts in a dataset. AFLITE is general and can be applied to any task and dataset. Our approach leverages a high capacity model to learn dataset specific biases on a small subset, then uses it to identify and filter artifact-prone instances in the remainder of the dataset to yield a final dataset that is possibly closer to the intended task.

We first evaluate the effectiveness of our method on synthetic data and show that AFLITE lowers the performance of models relying on annotation artifacts while preserving the performance of models whose representation captures the underlying tasks. In addition, while AFLITE aims to retain the more challenging, confusing instances, our experiments show that it can successfully remove biased instances that are adversarial to the correct representation of the data.

Finally, we apply the method to several benchmark datasets across various tasks and domains. In language understanding, we apply AFLITE to the SNLI (Bowman et al., 2015) and MNLI (Williams et al., 2018) datasets for natural language inference, and to QNLI (Wang et al., 2018) for question answering. We show a 30% absolute gap in performance in the current state-of-art methods before and after AFLITE. In computer vision, AFLITE reduces the performance of image classification neural methods on ImageNet (Russakovsky et al., 2015), showing a 49% absolute gap.

## 2 DATASET REDUCTION FOR REPRESENTATION-BIAS MINIMIZATION

In this section, we introduce AFLITE, a general approach for reducing the scope of bias in datasets. Large datasets run the risk of prioritizing performance on the data-rich *head* of the distribution, where examples are plentiful, and discounting the *tail*. Our goal is to minimize the ability of a model to exploit biases in the head of the distribution, while preserving the inherent complexity of the *tail*.

Let $\Phi$ represent a feature representation, defined over a dataset $\mathcal{D} = (X, Y)$. With AFLITE, we seek a subset $S \subset \mathcal{D}$ of size $|S| \geq n$ that is maximally resilient to the features uncovered by $\Phi$. For any identically-distributed train-test split of $\mathcal{D}$, the features extracted by $\Phi$ should not generalize to the held-out set. Our approach allows for any choice of feature representation.

**Formalization** More formally, let $\mathcal{M}$ denote a family of classification models (e.g., logistic regression, SVM, or a particular neural architecture) that can be trained on subsets $S$ of $D = (X, Y)$

using features $\Phi(X)$. We define the *representation bias of $\Phi$ in $S$ w.r.t $\mathcal{M}$*, denoted $\mathcal{R}(\Phi, S, \mathcal{M})$, as the best possible out-of-sample classification accuracy achievable by models in $\mathcal{M}$ when predicting the true labels $Y$ using features $\Phi(X)$. For a given target reduced dataset size of at least $n$, the goal is to find a subset $S \subset D, |S| \geq n$ that minimizes this representation bias in $S$ w.r.t. $\mathcal{M}$:

$$\min_{S \subset D, |S| \geq n} \mathcal{R}(\Phi, S, \mathcal{M}) \tag{1}$$

Eq. (1) corresponds to the optimum bias reduction, referred to as AFOPTIMUM. $\mathcal{R}(\Phi, S, \mathcal{M})$ can be formulated as the expected classification accuracy resulting from the following process. Let $q : 2^S \to [0, 1]$ be a probability distribution over subsets $T = (X^T, Y^T)$ of $D$. The process is to randomly choose a subset $T$ with probability $q(T)$, train a bias estimator $M_T \in \mathcal{M}$ on $D \setminus T$, and evaluate its classification accuracy $f_{M_T}(\Phi(X^T), Y^T)$ on $T$. Note that the resulting classification accuracy on $T$ itself is a random variable, since the training set $D \setminus T$ is random. We define the expected value of this classification accuracy to be the representation bias:

$$\mathcal{R}(\Phi, S, \mathcal{M}) \triangleq \mathbb{E}_{T \sim q} \left[ f_{M_T}(\Phi(X^T), Y^T) \right] \tag{2}$$

While this expression formalizes the intended objective function, it involves a large summation over subsets $T \subset S$ just to compute the representation bias present in a single set $S$. It does not suggest a practical way to compute the minimization in Eq. (1) without further considering each of the exponentially many subsets $S \subset D$ individually – thus an optimal solution for Equation (2) is intractable. To get around this difficulty, we reformulate the representation bias in $S$ as a sum factored over the $|S|$ individual instances $i \in S$. This will allow us to efficiently decide whether or not to include $i$ in the targeted, reduced subset we are constructing.

The idea is to aggregate the contribution of each $i$ towards the representation bias expression across all random choices of the training set $D \setminus T$. We call this the *predictability score $p(i)$* for $i$: on average, how reliably can the label $y_i$ be predicted using features $\Phi(x_i)$ when a model from $\mathcal{M}$ is trained on a randomly chosen training set $D \setminus T$ not containing $i$. The higher the value of $p(i)$, the easier it is to correctly classify the instance $(x_i, y_i)$ using model family $\mathcal{M}$. This is the signal we will use to decide whether to include $i$ in the reduced subset $S$ we are constructing.

With some abuse of notation, for $i \in D$, let $q(i) \triangleq \sum_{T \ni i} q(T)$ denote the marginal probability of choosing a subset $T$ that contains $i$. The ratio $\frac{q(T)}{q(i)}$ is then the probability of $T$ conditioned on it containing $i$. Let $f_{M_T}(\Phi(x_i), y_i)$ be the classification accuracy of $M_T$ on $i$. The reformulation of representation bias in terms of predictability scores of individual instances works as follows:

$$
\begin{aligned}
\mathbb{E}_{T \sim q} \left[ f_{M_T}(\Phi(X^T), Y^T) \right] &= \sum_{T \subset S} q(T) \cdot \frac{1}{|T|} \sum_{i \in T} f_{M_T}(\Phi(x_i), y_i) \\
&= \sum_{T \subset S} \sum_{i \in T} q(T) \cdot \frac{f_{M_T}(\Phi(x_i), y_i)}{|T|} \\
&= \sum_{i \in S} \sum_{\substack{T \subset S \\ T \ni i}} q(T) \cdot \frac{f_{M_T}(\Phi(x_i), y_i)}{|T|} \\
&= \sum_{i \in S} q(i) \sum_{\substack{T \subset S \\ T \ni i}} \frac{q(T)}{q(i)} \frac{f_{M_T}(\Phi(x_i), y_i)}{|T|} \\
&= \sum_{i \in S} q(i) \, \mathbb{E}_{T \subset S, \, T \ni i} \left[ \frac{f_{M_T}(\Phi(x_i), y_i)}{|T|} \right] \\
&= \sum_{i \in S} p(i)
\end{aligned}
$$

where $p(i)$ is the predictability score of $i$ defined as:

$$p(i) \triangleq q(i) \, \mathbb{E}_{T \subset S, \, T \ni i} \left[ \frac{f_{M_T}(\Phi(x_i), y_i)}{|T|} \right] \tag{3}$$

While the method works for any probability distribution $q$ with non-zero support on all samples, for simplicity of exposition, we restrict $q$ to be the uniform distribution over all subsets $T \subset S$ of a fixed size. This makes both $|T|$ and $q(i)$ fixed constants; in particular, $q(i) = \binom{|S|-1}{|T|-1}/\binom{|S|}{|T|} = \frac{|T|}{|S|}$. This reduces the predictability score expression of Eq. (3) to the simplified variant $\tilde{p}(i)$:

$$\tilde{p}(i) \triangleq \frac{1}{|S|} \mathbb{E}_{T \subset S, \, T \ni i} \left[ f_{M_T}(\Phi(x_i), y_i) \right] \tag{4}$$

Putting the pieces together, we have a factored reformulation of the representation bias in Eq. (2):

$$\mathcal{R}(\Phi, S, \mathcal{M}) = \sum_{i \in S} \tilde{p}(i) \tag{5}$$

Armed with this factored representation, we return to the task of identifying an $S \subset D, |S| \geq n$ that minimizes the representation bias. We use the simplified predictability scores (henceforth simply referred to as predictability scores) as a heuristic metric to decide which $i \in D$ to include in $S$. We consider three approaches that iteratively filter out the most predictable instances from $D$ to arrive at $S$. In all cases, we use a fixed training set size $|S \setminus T| = t < n$. Further, since a larger filtered set is generally desirable, we terminate the filtering process early (i.e., while $|S| > n$) if the predictability score for every $i$ falls below a pre-specified early stopping threshold $\tau \in [0, 1]$.

The three approaches are as follows. (A) A simple *greedy approach* starts with the full set $S = D$, identifies an $i \in S$ with the highest predictability score, removes it from $S$, and repeats up to $|D| - n$ times. (B) A *greedy slicing approach* identifies the instances with the $k$ highest predictability scores, removes all of them from $S$, and repeats the process up to $\lfloor \frac{|D|-n}{k} \rfloor$ times. (C) A *slice sampling approach* where, instead of greedily choosing the top $k$ instances, it randomly samples $k$ instances with probabilities proportional to their predictability scores.[1] In this paper, we use the greedy slicing approach in our experiments and refer to it as AFLITE. While the optimum bias reduction via AFOPTIMUM is intractable, its light-weight version, AFLITE, is applicable in practice.

The slice sampling approach can be efficiently implemented using what is known as the Gumbel method or Gumbel trick (Gumbel & Lieblein, 1954; Maddison et al., 2014), which uses random perturbations to turn sampling into a simpler problem of optimization. This has recently found success in several probabilistic inference applications (Kim et al., 2016; Jang et al., 2016; Maddison et al., 2016; Balog et al., 2017; Kool et al., 2019). Starting with the log-predictability scores $\log \tilde{p}(i)$ for various $i$, the idea is to perturb them by adding an independent random noise $\gamma_i$ drawn from the standard Gumbel distribution. Interestingly, the maximizer $i^*$ of $\gamma_i + \log \tilde{p}(i)$ turns out to be an exact sample drawn from the (unnormalized) distribution defined by $\tilde{p}$. Note that $i^*$ is a random variable since the $\gamma_i$ are drawn at random. This result can be generalized (Vieira, 2014) for slice sampling: the $k$ highest values of Gumbel-perturbed log-predictability scores correspond to sampling, without replacement, $k$ items from the probability distribution defined by $\tilde{p}$. The Gumbel method is typically applied to exponentially large combinatorial spaces, where it is challenging to scale up. In our setting, however, the overhead is minimal since the cost of drawing a random $\gamma_i$ is negligible compared to computing $\tilde{p}(i)$.

**Implementation** Algorithm 1 provides an implementation of AFLITE. The algorithm takes as input a dataset $D = (X, Y)$, a representation $\Phi(X)$ we are interested in minimizing the bias in, a model family $\mathcal{M}$ (e.g., linear classifiers), a target dataset size $n$, size $m$ of the support of the expectation in Eq. (4), training set size $t$ for the classifiers, size $k$ of each slice, and an early-stopping filtering threshold $\tau$. Importantly, for efficiency, $\Phi(X)$ is provided to AFLITE in the form of *precomputed* embeddings for all of $X$. To obtain $\Phi(X)$ in practice, we train a first model on a small fraction of the data based on the learning curve in low-data regime, and do not reuse this data for the rest of our experiments. Moreover, this fraction corresponds to the training size $t$ for AFLITE and it remains unchanged across iterations. We follow the iterative filtering approach, starting with $S = D$ and iteratively removing some instances with the highest predictability scores using the

---

[1] All of these approaches can be improved further by considering not only the predictability score of the top-$k$ instances $i$ but also (via retraining without these instances) how their removal would influence the predictability scores of other instances in the next step. We found our computationally lighter approaches to work well even without the additional overhead of such look-ahead.

---

**Algorithm 1:** AFLITE

---

**Input:** dataset $D = (X, Y)$, pre-computed representation $\Phi(X)$, model family $\mathcal{M}$, target dataset size $n$,
number of random partitions $m$, training set size $t < n$, slice size $k \leq n$, early-stopping threshold $\tau$

**Output:** reduced dataset $S$

1  $S = D$
2  **while** $|S| > n$ **do**
  // Filtering phase
3    **forall** $i \in S$ **do**
4      Initialize a multi-set of out-of-sample predictions $E(i) = \emptyset$
5    **for** *iteration* $j : 1..m$ **do**
6      Randomly partition $S$ into $(T_j, S \setminus T_j)$ s.t. $|S \setminus T_j| = t$
7      Train a classifier $\mathcal{L} \in \mathcal{M}$ on $\{(\Phi(x), y) \mid (x, y) \in S \setminus T_j\}$ ($\mathcal{L}$ is typically a linear classifier)
8      **forall** $i = (x, y) \in T_j$ **do**
9        Add the prediction $\mathcal{L}(\Phi(x))$ to $E(i)$
10   **forall** $i = (x, y) \in S$ **do**
11     Compute the predictability score $\tilde{p}(i) = |\{\hat{y} \in E(i) \ s.t. \ \hat{y} = y\}| / |E(i)|$
12   Select up to $k$ instances $S'$ in $S$ with the highest predictability scores subject to $\tilde{p}(i) \geq \tau$
13   $S = S \setminus S'$
14   **if** $|S'| < k$ **then**
15     **break**
16 **return** $S$

---

greedy slicing strategy. Slice size $k$ and number of partitions $m$ are determined by the available computation budget.

At each filtering phase, we train models (linear classifiers in our implementation) on $m$ different random partitions of the data, and collect their predictions on their corresponding test set. For each instance $i$, we compute its *predictability score* as the ratio of the number of times its label $y_i$ is predicted correctly, over the total number of predictions for it. We rank the instances according to their predictability score and use the greedy slicing strategy of removing the top-$k$ instances whose score is not less than the early-stopping threshold $\tau$. We repeat this process until fewer than $k$ instances pass the $\tau$ threshold in a filtering phase or fewer than $n$ instances remain.

## 3   EXPERIMENTAL ANALYSIS

We evaluate AFLITE across various domains (synthetic, natural language processing, computer vision), different tasks in a given domain (language inference and question answering in NLP), different datasets for a given task (SNLI and MNLI in natural language inference), and different representations for a given dataset (pre-computed embeddings from *ESIM+GLoVe*, *BERT*, *RoBERTa* for the SNLI dataset).

### 3.1   SYNTHETIC EXPERIMENTS

We demonstrate the utility of AFLITE in a synthetic data setting. Our dataset consists of two-dimensional data, arranged in concentric circles, at four different levels of separation, as shown in the Figure 2. As is evident, a linear function might not be adequate for separating the two classes; it requires a more complex non-linear model such as an SVM with an RBF kernel. [2]

We add class-specific artificially constructed features (artifacts) sampled from two different Gaussian distributions. These features are only added to $75\%$ of the data in each class, while for the rest of the data, we insert random (noise) features. These artifacts make the task solvable through a linear function. Furthermore, for the first dataset, with the largest separation, we flipped the labels of some examples with artifacts, making the data slightly adversarial even to the RBF. Both models can clearly leverage the artifacts, and demonstrate improved performance over a baseline without artifacts.

---

[2]We use standard implementations from scikit-learn: `https://scikit-learn.org/stable/`.

| Model | $D$ | $D_{92k}$ | $D(\Phi_{ESIM+GLoVe})$ | $D(\Phi_{BERT})$ | $D(\Phi_{RoBERTa})$ |
|---|---|---|---|---|---|
| ESIM+ELMo (Peters et al., 2018) | 88.7 | 86.0 | 61.5 | 54.2 | 51.9 |
| BERT (Devlin et al., 2019) | 91.3 | 87.6 | 74.7 | 61.8 | 57.0 |
| RoBERTa (Liu et al., 2019b) | 92.6 | 88.3 | 78.9 | 71.4 | 62.6 |
| Max-PPMI baseline | 54.5 | 52.0 | 41.1 | 41.5 | 41.9 |
| BERT-*HypOnly* | 71.5 | 70.1 | 52.3 | 46.4 | 48.4 |
| RoBERTa-*HypOnly* | 72.0 | 70.4 | 53.6 | 49.5 | 48.5 |
| *Human performance* | 88.1 | 88.1 | 82.3 | 80.3 | 77.8 |
| *Training set size* | 550k | 92k | 138k | 109k | 92k |

Table 1: *Dev* accuracy (%) on the original SNLI dataset $D$ and the datasets obtained through various representation-bias minimization. The *-HypOnly* baselines correspond to models trained on the instances restricted to their hypotheses.

Once we apply AFLITE, as expected, the number of examples with artifacts is reduced considerably, making the task hard once again for the linear model, but still solvable for the non-linear one. The filtered dataset is shown in the bottom half of Fig. 2, and the captions indicate the performance of a linear and an SVM model. For the first dataset, we see that AFLITE removes most of those examples with flipped labels.

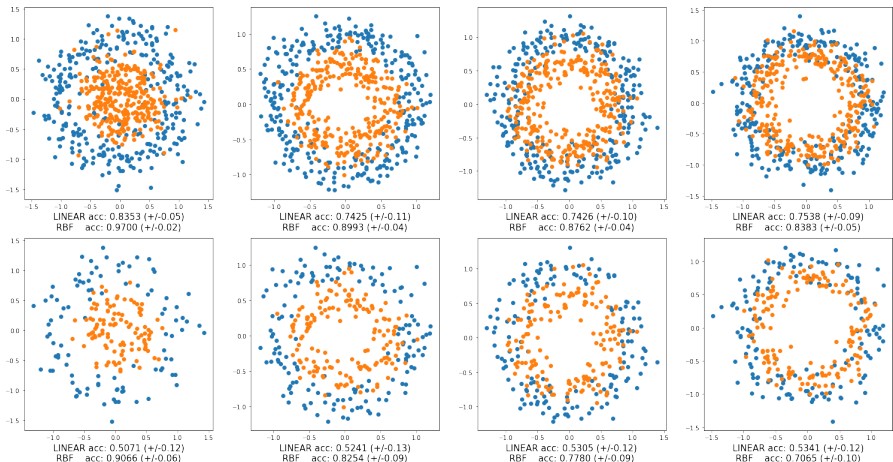

Figure 2: Four sample datasets with artifacts as input to AFLITE (top). Blue and orange indicate two different classes. Only the original two dimensions are shown, not the artifacts. For the leftmost dataset with the highest separation, we flip some labels at random, so even an RBF kernel cannot achieve perfect performance. AFLITE makes the data more challenging for the models (bottom).

## 3.2 NLP EXPERIMENTS

We evaluate AFLITE on two NLP tasks, namely NLI and question answer sentence selection. We use two popular NLI large-scale datasets – SNLI (Bowman et al., 2015) and MNLI (Wang et al., 2018). For the answer sentence selection task, we use QNLI which is a transformed version of the SQuAD question answering dataset (Rajpurkar et al., 2016) converted to binary classification where systems determine whether a sentence contains the answer to a question.

**SNLI** Each instance in the SNLI dataset consists of a premise-hypothesis pair that belongs to one out of three possible categories (*entailment*, *contradiction*, or *neutral*) based on the relationship between the premise and the hypothesis.

For SNLI, we experiment with three different feature representations derived from strong baseline models: $\Phi_{BERT}$ and $\Phi_{RoBERTa}$ which are based on BERT (Devlin et al., 2019) and RoBERTa (Liu et al., 2019b), large-scale pretrained masked language models, plus $\Phi_{ESIM+GLoVe}$ which uses the

| Model | MNLI | | QNLI | |
|---|---|---|---|---|
| | $D$ | $D(\Phi_{RoBERTa})$ | $D$ | $D(\Phi_{RoBERTa})$ |
| BERT (Devlin et al., 2019) | 86.6 | 55.8 | 92.0 | 63.5 |
| RoBERTa (Liu et al., 2019b) | 90.3 | 66.2 | 93.7 | 77.7 |
| BERT-*PartialInput* | 59.7 | 43.2 | 62.6 | 56.6 |
| RoBERTa-*PartialInput* | 60.3 | 44.4 | 63.9 | 59.4 |

Table 2: *Dev* accuracy (%) on the original MNLI-matched and QNLI datasets and the datasets obtained through $\Phi_{RoBERTa}$-representation-bias minimization. The *-PartialInput* baselines correspond to models trained on partial, incomplete input, namely the *Hypotheses* for MNLI instances and the *Answers* for QNLI instances.

ESIM model (Chen et al., 2016) with GLoVe word embeddings (Pennington et al., 2014). In all cases, feature representation $\Phi$ is trained on a random sample of 10% of the original training instances, and feature representations are extracted from the final layer before the output layer. These features are pre-computed for all remaining instances while we discard the instances (10% of training) used for training the embeddings in the subsequent steps of our algorithm. Additionally, to measure the ability of a weaker adversary to filter biases only learned by a stronger model, we evaluate the filtered datasets (for SNLI) with three different models: (i) ESIM with ELMo embeddings (Peters et al., 2018), (ii) BERT-large, and (iii) RoBERTa-large models.

Table 1 shows the results for SNLI. In all cases, applying AFLITE substantially reduces overall model accuracy, with typical drops of 15-35% depending on the models used for learning the feature representations and those used for evaluation of the filtered dataset. In general, performance is lowest when using the strongest model (RoBERTa) for learning feature representations. Results also highlight the ability of weaker adversaries to produce datasets that are still challenging for much stronger models with a drop of 13.7% for RoBERTa using $\Phi_{ESIM+GLoVe}$ as feature representation. We also include a model that uses Point-wise Mutual Information (PMI) between words in a given instance and the target label as a feature. The baseline captures the extent to which datasets exhibit word-association artifacts. While this baseline is relatively weaker than other models, we still show that its performance reduce from 54.5% on $D$ to 41.9% on the $D(\phi_{RoBERTa})$ dataset.

It might seem unsurprising that reducing the size of the training set results in lower performance. To control for the confounding factor of the dataset size, we create another filtered dataset $D_{92k}$, sampled randomly from $D$ such that its size is approximately equal to the size of $D(\phi_{RoBERTa})$ dataset. All models achieve nearly the same performance as their performance on the full dataset – even when trained on just one-fifth the original dataset size. This result further points to the fact that current benchmark datasets contain significant redundancy within its instances.

Finally, to demonstrate the value of the iterative, ensemble-based AFLITE algorithm, we compare with a baseline where using a single model, we filter out the most predictable examples in a single iteration — a non-iterative, single-model version of AFLITE. A RoBERTa-large model trained on this subset (of the same size as $D(\phi_{RoBERTa})$) achieves a dev accuracy of 72.1%. Compared to the performance of RoBERTa on $D(\phi_{RoBERTa})$ (62.6%, see Table 1), it makes this baseline a sensible yet less effective approach. In particular, this illustrates the need for an iterative procedure involving models trained on multiple partitions of the remaining data in each iteration.

We also report the k-nearest neighbors distances between examples in the train and heldout data in Table 3. We consider distances for examples within each class, as well as examples across classes. The distances are computed using cosine similarity between pooled features from BERT-based model (features for the [CLS] token, indicating a sentence-pair feature) trained on the original SNLI dataset. Distances are measured between samples from the heldout data, and their nearest neighbors in the training data, before and after filtering. Distances generally increase after filtering, indicating that AFLITE promotes selecting a diverse set of examples from the dataset. The only exception to the rule is the neutral class, where distances to other classes decrease – this is not surprising since the neutral class is known to be associated with the least number of artifacts (Gururangan et al., 2018).

|  | Before AFLITE | | | | After AFLITE | | | |
|---|---|---|---|---|---|---|---|---|
|  | Top1 | Top5 | Top10 | Top50 | Top1 | Top5 | Top10 | Top50 |
| Entailment | 0.29 | 1.32 | 2.45 | 9.5 | 0.32 | 1.42 | 2.64 | 9.98 |
| Neutral | 0.40 | 1.89 | 3.68 | 16.83 | 0.42 | 1.97 | 3.80 | 16.66 |
| Contradiction | 0.49 | 2.41 | 4.77 | 23.09 | 0.52 | 2.49 | 4.84 | 22.10 |
| Entailment vs others | 0.32 | 1.48 | 2.87 | 12.98 | 0.34 | 1.53 | 2.92 | 12.31 |
| Neutral vs others | 0.43 | 2.05 | 3.99 | 18.42 | 0.41 | 1.94 | 3.72 | 16.41 |
| Contradiction vs others | 0.49 | 2.38 | 4.70 | 22.61 | 0.53 | 2.51 | 4.87 | 22.38 |

Table 3: KNN-distances by class, before and after applying (RoBERTa-filtered) AFLITE to SNLI.

| Model | HANS | | | | NLI-Diagnostics | | | Adversarial-NLI | | |
|---|---|---|---|---|---|---|---|---|---|---|
|  | *All* | *Lex.* | *Subseq.* | *Constit.* | *All* | *Logic* | *Knowl.* | *Rd1* | *Rd2* | *Rd3* |
| RoBERTa | 70.7 | 84.4 | 35.4 | 13.4 | 59.3 | 52.8 | 48.9 | 58.5 | 48.3 | 50.1 |
| RoBERTa-AFlite | **74.5** | **96.3** | **56.6** | **57.4** | **62.0** | **53.2** | **57.7** | **65.1** | **49.1** | **52.8** |

Table 4: SNLI accuracy (%) on three out-of-distribution evaluation tasks, comparing RoBERTa-large models pre-trained on the original SNLI data, and on AFLITE-filtered data. On the HANS dataset, both models are evaluated on *All*, as well as on the non-entailment cases of the three syntactic heuristics (*Lexical overlap*, *Subsequence*, and *Constituent*). The NLI-Diagnostics dataset is broken down into the full dataset (*All*), as well as the instances requiring logical reasoning (*Logic*) and the ones requiring world and commonsense knowledge (*Knowledge*). For Adversarial NLI, we finetuned both models on the in-distribution training data for each round (*Rd1*, *Rd2*, and *Rd3*).

**MNLI and QNLI** Following the same procedure described above, we apply AFLITE on the MNLI and QNLI datasets. Since RoBERTa resulted in the largest drops in performance across the board in SNLI, we only experiment with RoBERTa as adversary for MNLI and QNLI. While RoBERTa achieves over $90\%$ on both original datasets, its performance drops to $66.2\%$ for MNLI and to $77.7\%$ for QNLI on the reduced datasets. Similarly, partial input baseline performance also decreases substantially on both dataset compared to their performance on the original dataset. Table 2 shows these results. We show that AFLITE consistently result in reduced accuracy on the filtered datasets across multiple NLP benchmark datasets, even after controlling for the size of the training set.

### 3.2.1 OUT-OF-DISTRIBUTION NLI

We measure the performance of AFLITE on three other benchmarks for NLI evaluation, which provide out-of-distribution examples to challenge reliance on dataset biases in the original SNLI data (Glockner et al., 2018; Naik et al., 2018). Such benchmarks approximate the performance of NLI models in the wild. NLI Diagnostics (Wang et al., 2018) is a set of hand-crafted examples designed to demonstrate model performance on several fine-grained semantic categories, such as logical reasoning and commonsense knowledge. HANS (McCoy et al., 2019) contains evaluation examples designed to avoid common structural heuristics (such as word overlap) which could be used by models to correctly predict NLI inputs, without true inferential reasoning. Adversarial NLI (Nie et al., 2019) consists of premises collected from Wikipedia and other news corpora, and human generated hypotheses, arranged at different tiers of the challenge they present to a model, using a human and model in-the-loop procedure. Given that these benchmarks are collected independently of the original SNLI task, the biases from SNLI are less likely to carry over; however these benchmarks might contain their own biases (Liu et al., 2019a).

AFLITE assigns a predictability score to all samples in a dataset, resulting in an ordering of the data. Filtering out examples from the head of the data distribution based on this order yields more accurate benchmarks for measuring true model performance. On the other hand, transferability to out-of-distribution data would involve a greater balance between examples from the head and the tail ends of the data distribution. Hence we evaluate AFLITE for generalization using a larger filtered subset, amounting to only a third of the full training data. We present these results on all the above benchmarks in Table 4. On the two diagnostic datasets (HANS and NLI-Diagnostics), we perform a zero-shot evaluation of the two models. Adversarial NLI allows to test for transfer

| Model | 100% Train, Original Val | | 20% Train, Original Val | | AFLITE | |
|---|---|---|---|---|---|---|
| | Top-1 | Top-5 | Top-1 | Top-5 | Top-1 | Top-5 |
| EfficientNet-B0 | 76.3 | 93.2 | 58.5 | 81.2 | 18.1 | 48.1 |
| EfficientNet-B2 | 79.8 | 94.9 | 60.9 | 82.8 | 20.7 | 53.1 |
| EfficientNet-B4 | 82.6 | 96.3 | 64.4 | 85.8 | 23.3 | 58.8 |
| EfficientNet-B7 | **84.4** | **97.1** | **73.8** | **90.8** | **24.5** | **60.6** |
| ResNet-34 | 78.4 | 94.4 | 51.8 | 74.3 | 11.1 | 30.2 |
| ResNet-50 | 79.2 | 94.7 | 53.2 | 75.5 | 12.2 | 30.2 |
| ResNet-101 | 80.1 | 95.4 | 55.6 | 77.5 | 12.3 | 32.1 |
| ResNet-152 | 80.6 | 95.5 | 56.5 | 78.2 | 13.2 | 33.8 |

Table 5: Experimental results on ImageNet. We compare between three settings: the original dataset's train-test splits, using 20% of the training set but evaluating on the validation set, and using the AFLITE produced training and validation sets. AFLITE produces a training dataset that is also 20% of the training set size, making it a fair comparison in terms of dataset examples. The results show a significant drop in Top-1 and Top-5 accuracy: the Top-1 accuracy goes down by roughly 40 percentage points per model in this new training and evaluation setting.

capabilities, by finetuning these models on each of the three training datasets (*Rd1*, *Rd2* and *Rd3*). On each of the benchmarks above, the model trained on the AFLITE data consistently outperforms the model trained on the full SNLI data. challenging examples in the HANS benchmark, which targets models purely relying on lexical and syntactic cues. Similarly, our model performs better on the instances in NLI-Diagnostics that require logical reasoning and commonsense knowledge, as opposed to instances that can be solved through lexical entailment alone.

## 3.3 IMAGENET EXPERIMENTS

We evaluate AFLITE on image classification through ImageNet (ILSVRC2012) classification. On ImageNet, we use the state-of-the-art EfficientNet-B7 model as our core feature extractor $\Phi$ (Tan & Le, 2019). The EfficientNet model is learned from scratch on a fixed 20% sample of the ImageNet training set, using AutoAugment data augmentation (Cubuk et al., 2019). We then use the 2560-dimensional features extracted by EfficientNet-B7 as then underlying representation for AFLITE to use to filter the remaining dataset.

In Table 5, we evaluate the robustness of the filtered dataset by considering ImageNet accuracy across the EfficientNet and ResNet model families (He et al., 2016). When lowering the size of the training set – down to 20% of the original, we find a large drop in performance. The Efficient-Net models seem to suffer less – from 84% to 73% on EfficientNet-B7 versus 80.6% to 56.5% on ResNet-152. However, the biggest performance drop comes from training and evaluating on the AFLITE-filtered dataset: the top performer is still EfficientNet-B7, but its accuracy drops to 24.5% top-1. This is despite controlling for dataset size, as well as discrepancy between the training and validation sets.

Overall, these results suggest that image classification – even within a subset of the closed world of ImageNet – is far from solved. These results echo other findings that suggest that common biases that naturally occur in web-scale image data, such as towards canonical poses (Alcorn et al., 2019) or towards texture rather than shape (Geirhos et al., 2018), are problems for ImageNet-trained classifiers. Indeed, the randomly-selected ImageNet images in Figure 1 suggest that the AFLITE algorithm learns to identify subsets of the data that are particularly challenging.

## 4 RELATED WORK

Our proposed framework for artifact reduction is related to the adversarial filtering (AF) algorithm in Zellers et al. (2018), yet distinct in two key ways: our approach is (i) much more broadly applicable (by not requiring over generation of data instances), and (ii) considerably more lightweight (by not requiring re-training a model at each iteration of AF). Variants of this AF approach have recently been used to create other datasets such as HellaSwag (Zellers et al., 2019) and ANLI (Bhagavatula

et al., 2019) by iteratively perturbing dataset instances until a target model cannot fit the resulting dataset. While effective, these approaches run into three main pitfalls. First, dataset curators need to explicitly devise a strategy of collecting or generating perturbations of a given instance. Second, the approach runs the risk of distributional bias where a discriminator can learn to distinguish between machine generated instances and human-generated ones. Finally it requires re-training a model at each iteration, which is computationally expensive especially when using a large model such as BERT (Devlin et al., 2019) as the adversary. In contrast, AFLITE focuses on addressing dataset biases from existing datasets instead of adversarially perturbing instances. AFLITE was earlier proposed by Sakaguchi et al. (2019) to create the Winogrande dataset. This paper presents more thorough experiments, theoretical justification and results from generalizing the proposed approach to multiple popular NLP and Vision datasets.

AFLITE is also inspired by Gururangan et al. (2018), who study lexical biased prevalent in the SNLI dataset (Bowman et al., 2015) and use point-wise mutual information (PMI) between a word and an inference class to determine the words that are highly indicative of the target label. Instead of lexical features, we adopt a deeper representation of the instances using their *pre-computed* dense feature representations. We use an ensemble of linear classifiers trained on random subsets of the data to determine whether the dense feature representations are highly indicative of the target label. If so, we discard the corresponding instances and proceed iteratively.

Li & Vasconcelos (2019) recently proposed REPAIR, a method to remove representation bias by dataset resampling. While resampling is a common technique for balancing datasets, the motivation in REPAIR is to learn a probability distribution over the dataset that favors instances that are hard for a given representation. This approach targets how to train better, less-biased models as opposed to creating datasets with fewer artifacts. In addition, the implementation of REPAIR relies on in-training classification loss as opposed to out-of-sample generalization accuracy. RESOUND (Li et al., 2018) is another method that quantifies the representation biases of datasets. It uses the representation biases to assemble a new K-class dataset with smaller biases by sampling an existing C-class dataset ($C > K$).

Arjovsky et al. (2019) argue that unstable, spurious correlations in the data would generalize poorly to novel test environments. Thus, they propose Invariant Risk Minimization as an objective that promotes learning representations of the data which are stable across environments. Instead of learning optimal classifiers, our aim is to remove instances that exhibit artifacts in a dataset.

## 5 CONCLUSION

We presented AFLITE – a novel iterative greedy algorithm that adversarially filters out data points to arrive at a reduced dataset with more realistic problem distributions and considerably fewer spurious biases. We apply AFLITE to four widely-used datasets, including SNLI and ImageNet, where reported performance is extremely high – and show that state-of-the-art performance on the resulting filtered dataset drops by 30 points for SNLI and drops from 84.4% to 24.5% Top-1 accuracy for ImageNet. In extensive analysis we show that AFLITE is effective on real as well as synthetic datasets. We hope that dataset creators will employ AFLITE to identify unobservable artifacts before releasing new challenge datasets for the research community in order to have a more reliable estimate of model performance on future AI benchmarks.

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
