# OpenReview forum: "Adversarial Filters of Dataset Biases"
_ICLR.cc/2020/Conference — Reject_

### Official Review · AnonReviewer3 · 2019-10-17
**Official Blind Review #3**

**Rating:** 6

**Review:**

This paper proposes to learn a subset of a given dataset that acts as an adversary, that hurts the model performance when used as a training dataset. The central claim of the paper is that existing datasets on which models are trained are potentially biased, and are not reflective of real world scenarios. By discarding samples that add to this bias, the idea is to make the model perform better in the wild. The authors propose a method to do so, and then refine it so that the resulting solution is tractable. They implement the method on several datasets and show that by finding these adversarial samples, they indeed hurt model performance.

COMMENTS:
- Overall the method seems to be something like what is done in k-fold CV, except here we want to find a subset that is the worst at predicting model performance. To this end, I find the introduction of terms like "representation bias" and "predictability scores" unnecessary. Why not model the entire problem in terms of classification error?

- Page 3 : the last 2 paragraphs are repeated from above.

E: i read the author responses and they addressed my concern about model performance in the wild. I have updated my score to reflect this.

- eqn (3) and the set of equations above: for the math to work, you need q(i) to have non-zero support on all samples. To that end, the sentence that says it works for "any" q() is incorrect.

- The experiments back your claim that your method makes the data more challenging to train on. But that does not address the central idea, that the resultant models do better in the wild. If the aim is to make the models robust to real world, you have provided no evidence that your method does so.

- Table 1: the D_92k column is good comparison to have. Thanks.



**Experience Assessment:**

I have read many papers in this area.

**Review Assessment: Checking Correctness Of Derivations And Theory:**

I carefully checked the derivations and theory.

**Review Assessment: Checking Correctness Of Experiments:**

I assessed the sensibility of the experiments.

**Review Assessment: Thoroughness In Paper Reading:**

I read the paper at least twice and used my best judgement in assessing the paper.

---

> ### Author Response · Authors · 2019-11-14
> **Models trained on AFLite-filtered data indeed perform better in the wild**
>
> We thank Reviewer 3 for their comments, and address their main points below.
>
> AFLite and generalization:
> We now have evidence suggesting resultant models trained on AFLite-filtered data indeed perform better in the wild. Our new results on three additional benchmarks show that a model trained on the AFLite-filtered dataset generalizes better than a model trained on the full SNLI dataset (Sec 3.2.1; please also see overall response https://openreview.net/forum?id=H1g8p1BYvS&noteId=rJeVCr85sS ).
>
> Terminology:
> The notion of predictability score is indeed related to classification error and its empirical estimate is indeed reminiscent of k-fold cross-validation. Yet, to avoid confusion with the term classification error, we use the term “predictability score of an instance” for the out-of-sample classification accuracy averaged over several models trained on a number of random subsets of the data.
>
> Others:
> We have updated the contents of Page 3 to remove the repeated paragraphs, and have changed the description of q() to solely distributions with a non-zero support.

---

### Official Review · AnonReviewer1 · 2019-10-22
**Official Blind Review #1**

**Rating:** 6

**Review:**

Summary: This paper hypothesizes that even though we are able to achieve very impressive performance on benchmark datasets as of now (e.g. image net), it might be due to the fact that benchmarks themselves have biases. They introduce an algorithm that selects more representative data points from the dataset that allow to get a better estimate of the performance in the wild. The algorithm ends up selecting more difficult/confusing instances.

This paper is easy to read and follow (apart from some hickup with a copy of three paragraphs), but in my opinion of limited use/impact.

Comments:
1) There is a repetition of the " while this expression formalizes.." paragraph and the next paragraph and the paragraph "As opposed to .." is out of place. Please fix
2) I am not sure
- What applications the authors suggest. They seem to say that benchmark authors should run their algorithm and make benchmarks harder. To me it seems that benchmarks become harder because you remove most important instances from the training data (so Table 4 is not surprising - you remove the most representative instances so the model can't learn)
- how practically feasible it is.  Even if in previous point I am wrong, the algo requires retraining the models on subsets (m iterations). How large is this m?
3) Other potential considerations:
-  When you change the training size, the model potentially needs to be re-tuned (regularization etc) (although it might be not that severe since the size of the training data is preserved at t)
- How do u chose the values of hyperparams (t, m,k eta), how is performance of your algorithm depends on it
4) I don't see any good baselines to compare with - what if i just chose instances that get the highest prediction score on a model and remove these. How would that do? For NLP (SNLI) task i think this would be a more reasonable baseline than just randomly dropping the instances,
5) I wonder if you actually retrain the features after creating filtered dataset, new representation would be able to recover the performance.

I read authors rebuttal and new experiments that show that the models trained on filtered data generalize better are proving the point, thanks. Changing to weak accept


**Experience Assessment:**

I do not know much about this area.

**Review Assessment: Checking Correctness Of Derivations And Theory:**

I assessed the sensibility of the derivations and theory.

**Review Assessment: Checking Correctness Of Experiments:**

I carefully checked the experiments.

**Review Assessment: Thoroughness In Paper Reading:**

I read the paper at least twice and used my best judgement in assessing the paper.

---

> ### Author Response · Authors · 2019-11-14
> **AFLite recalibrates benchmarks and promotes generalization**
>
> We thank Reviewer 1 for their comments, and address each of the concerns below:
>
> Impact 1: AFLite recalibrates benchmarks:
> [R1 (2)] We would like to clarify that the goal of this work is to recalibrate benchmarks, for the purpose of reporting true model performance. This does not necessarily involve removing the most important instances from the training data, but those which are spuriously correlated with the ground truth, making the overall task trivially easier. Once such correlations are minimized (i.e. after filtering with AFLite), the performance reduces.
>
> Impact 2: Do models trained on AFLite-filtered data generalize better? YES! :
> A second contribution of our work is demonstrating that the presence of instances with spurious correlations with the ground truth prevents models from generalizing to real world data. Our new results on three additional benchmarks show that a model trained on the AFLite-filtered dataset generalizes better than a model trained on the full SNLI dataset (Sec 3.2.1; also please see overall response: https://openreview.net/forum?id=H1g8p1BYvS&noteId=rJeVCr85sS ).
>
> Computational Overhead:
> [R1 (2)] Our approach operates on precomputed representations of the instances and relies on inexpensive logistic regressions. As a result, it is very efficient, scalable and parallelizable. It can efficiently run on CPU machines as well, and an effective value for m is a multiple of the number of available cores (e.g., 64 or 128).
>
> Hyperparameters:
> [R1 (3)] We have updated the draft to provide more information about the different hyperparameters (Sec 2; Implementation). These were selected based on the learning curves observed when training the model that we use to generate feature embeddings for the rest of the data, as well as the available computational budget (#CPU cores etc.). The training size, t, is kept constant throughout the algorithm, as R1 correctly points out; hence we do not modify the hyperparameters across iterations.
>
> New Baselines:
> [R1 (4)] As Reviewer 1 suggested, we have now provided a baseline (Sec 3.2 paragraph 6) which filters out the most predictable examples in a single pass. This corresponds to a non-iterative version of AFLite. For the SNLI task, this baseline (dev acc = 72.1%), however, is not as powerful as the full AFLite model (dev acc = 62.6%). This demonstrates the need for an iterative procedure involving models trained on multiple partitions in each iteration.
>
> Retraining post-AFLite:
> [R1 (5)] We do indeed completely retrain models after creating the filtered dataset. With the exception of finetuning the same pretrained representations such as RoBERTa (publicly available), there is no sharing of parameters between the older model trained on the full dataset, and the newer model trained on the filtered dataset. Moreover, the parameters used during the AFLite filtering are not reused when reporting benchmark performance. As shown in our experiments, these new representations, however, are unable to recover the original performance.
>
> Others:
> [R1 (1)] We apologize for the duplicates, and have fixed the issue in the updated draft.

---

### Official Review · AnonReviewer2 · 2019-10-24
**Official Blind Review #2**

**Rating:** 6

**Review:**

the paper proposes an algorithm that adversarially filters out examples to reduce dataset-specific spurious bias. the key intuition is that the datasets are curated in a way that easy to obtain samples have higher probability to be admitted to the dataset. however, not all real world samples are easy to obtain. in other words, real world samples may follow a completely different distribution than curated samples with easy-to-obtain ones.

the proposed approach discounts the data-rich head of the datasets and emphasizes the data-low tail. they quantify data-rich / data-low by the best possible out-of-sample classification accuracy achievable by models when predicting.

then adjust the dataset via the expected out-of-sample classification accuracy. the idea of the paper is interesting and the experiments show a substantial reduction in the performance of existing algorithms. this make the paper a promising proposal.


**Experience Assessment:**

I do not know much about this area.

**Review Assessment: Checking Correctness Of Derivations And Theory:**

I assessed the sensibility of the derivations and theory.

**Review Assessment: Checking Correctness Of Experiments:**

I assessed the sensibility of the experiments.

**Review Assessment: Thoroughness In Paper Reading:**

I read the paper at least twice and used my best judgement in assessing the paper.

---

> ### Author Response · Authors · 2019-11-14
> **Thank you for the positive feedback!**
>
> We thank Reviewer 2 for their positive feedback. Please also see our overall response for some new experimental evidence explicitly addressing the robustness of models trained on AFLite-filtered data:  https://openreview.net/forum?id=H1g8p1BYvS&noteId=rJeVCr85sS

---

### Author Response · Authors · 2019-11-14
**Overall Comments (For All Reviewers)**

We thank the reviewers for their helpful comments.

Our AFLite algorithm filters instances exhibiting spurious correlations with the gold labels, from several popular benchmarks, suggesting that the benchmarks have collection biases [as noted by R1,R2]. Training on the filtered subset prevents models from overfitting to such correlations, hence yields decreased benchmark performance [as noted by R3]. This, in itself, is an important finding because a significant fraction of empirical, applied ML research is evaluated off of these benchmarks.

In addition, in response to R1 and R3, we have now explicitly tried to address this question:
- Do models trained on AF-Lite filtered data generalize better to data in the wild?

We found the answer to be “yes”! More concretely, we provide new experimental results on three additional benchmarks: zero-shot evaluation on two diagnostic Natural Language Inference (NLI) datasets  (HANS; McCoy et al., 2019, NLI Diagnostics; Wang et al., 2018) as well as transfer learning on the newly released Adversarial-NLI dataset (Nie et al., 2019). We obtain the following results using a RoBERTa model, with all results in accuracy (also in our updated Section 3.2.1):

                                                                           HANS
Dataset                              All        Lex.     Subsequence   Constituent
100% of SNLI               70.7%    84.4%                   35.4%             13.4%
AFLite-filtered SNLI    74.5%    96.3%                   56.6%             57.4%

                                                        NLI Diagnostics
Dataset                                   All       Logic    Knowledge
100% of SNLI                    59.3%      52.8%             48.9%
AFLite-filtered SNLI         62.0%      53.2%             57.7%

                                                      Adversarial NLI
Dataset                                   Rd1         Rd2       Rd3
100% of SNLI                      58.5%     48.3%    50.1%
AFLite-filtered SNLI           65.1%     49.1%    52.8%

In particular, our model is very robust to the challenging examples in the HANS benchmark (upto 44% improvement), which is aimed at confusing models purely relying on simple linguistic constructions in the input. These results are very encouraging, considering that the AFLite-filtered data is a small subset of the original data. Moreover, we can infer that the AFLite-filtered distribution is closer to the real-world data distribution - training on just the AFLite-filtered data produces a more robust model than training on the entire dataset.

The reviewers also brought up other helpful changes and suggestions, we have addressed those in responses to each reviewer.

---

### Author Response · Authors · 2020-08-05
**Updated version (Accepted at ICML 2020)**

We have updated the version of our paper with a version that is now in the proceedings of ICML 2020. Also found here: https://arxiv.org/abs/2002.04108

Note that the changes from the ICLR submission include experiments demonstrating the ability of models trained on AFLite-filtered data to generalize to out-of-distribution tasks in both NLP as well as vision. We also include detailed information about the choice of hyperparameters.

---

### Decision · Program_Chairs · 2019-12-19

**Decision:**

Reject

**Comment:**

This paper proposes to address the issue of biases and artifacts in benchmark datasets through the use of adversarial filtering. That is, removing training and test examples that a baseline model or ensemble gets wright.

The paper is borderline, and could have flipped to an accept if the target acceptance rate for the conference were a bit higher. All three reviewers ultimately voted weakly in favor of it, especially after the addition of the new out-of-domain generalization results. However, reviewers found it confusing in places, and R2 wasn't fully convinced that this should be applied in the settings the authors suggest. This paper raises some interesting and controversial points, but after some private discussion, there wasn't a clear consensus that publishing it as is would do more good than harm.